# Comparative Analysis of Acanthopanacis Cortex and Periplocae Cortex Using an Electronic Nose and Gas Chromatography–Mass Spectrometry Coupled with Multivariate Statistical Analysis

**DOI:** 10.3390/molecules27248964

**Published:** 2022-12-16

**Authors:** Li Sun, Jing Wu, Kang Wang, Tiantian Liang, Quanhui Liu, Junfeng Yan, Ying Yang, Ke Qiao, Sui Ma, Di Wang

**Affiliations:** Xiangyang Public Inspection and Testing Center, Xiangyang 441000, China

**Keywords:** Acanthopanacis Cortex, Periplocae Cortex, sensors, electronic nose, gas chromatography-mass spectrometry, Chinese Herbal Medicines, odor identification, multivariate statistical analysis

## Abstract

Chinese Herbal Medicines (CHMs) can be identified by experts according to their odors. However, the identification of these medicines is subjective and requires long-term experience. The samples of Acanthopanacis Cortex and Periplocae Cortex used were dried cortexes, which are often confused in the market due to their similar appearance, but their chemical composition and odor are different. The clinical use of the two herbs is different, but the phenomenon of being confused with each other often occurs. Therefore, we used an electronic nose (E-nose) to explore the differences in odor information between the two species for fast and robust discrimination, in order to provide a scientific basis for avoiding confusion and misuse in the process of production, circulation and clinical use. In this study, the odor and volatile components of these two medicinal materials were detected by the E-nose and by gas chromatography–mass spectrometry (GC-MS), respectively. An E-nose combined with pattern analysis methods such as principal component analysis (PCA) and partial least squares (PLS) was used to discriminate the cortex samples. The E-nose was used to determine the odors of the samples and enable rapid differentiation of Acanthopanacis Cortex and Periplocae Cortex. GC-MS was utilized to reveal the differences between the volatile constituents of Acanthopanacis Cortex and Periplocae Cortex. In all, 82 components including 9 co-contained components were extracted by chromatographic peak integration and matching, and 24 constituents could be used as chemical markers to distinguish these two species. The E-nose detection technology is able to discriminate between Acanthopanacis Cortex and Periplocae Cortex, with GC-MS providing support to determine the material basis of the E-nose sensors’ response. The proposed method is rapid, simple, eco-friendly and can successfully differentiate these two medicinal materials by their odors. It can be applied to quality control links such as online detection, and also provide reference for the establishment of other rapid detection methods. The further development and utilization of this technology is conducive to the further supervision of the quality of CHMs and the healthy development of the industry.

## 1. Introduction

Acanthopanacis Cortex has been used in clinical application for a long time in China. The root bark of *Acanthopanax gracilistylus* W. W. Smith (Araliaceae) is used in Chinese Herbal Medicines (CHMs) to expel wind-dampness, tonify the liver and kidneys, and strengthen muscles and bones; its clinical use was first recorded in the text “Shen Nong Ben Cao Jing”. Nowadays, the Chinese Pharmacopoeia has embodied the root bark of *A. gracilistylus* as the qualified resource of Acanthopanacis Cortex. To date, Acanthopanacis Cortex has been reported to exhibit anti-inflammatory [1] and potential anti-tumor activities [2], as well as demonstrate therapeutic effects for postmenopausal osteoporosis [3]. As a candidate therapy, Acanthopanacis Cortex endowed significant protection against GalN/LPS-induced lethality, thereby showing potential treatment for fulminant hepatitis [4]. There is a long history of over two thousand years for the preparation of Chinese medicinal liquors (CML), officially called health-care liquors, which are made of alcohol and a large variety of traditional Chinese medicines (TCMs). Wujiapi liquor is one of the famous Chinese medicinal liquors and has been produced for hundreds of years in Southern China. Acanthopanax gracilistylus wine, which is made of Acanthopanacis Cortex and other herbs soaked in liquor, is popularly used as a health supplement product to treat rheumatic arthritis because of its clinical effects [5].

However, in most medical material markets, Acanthopanacis Cortex is often confused easily with other herbs, thereby causing potential safety issues. Acanthopanacis Cortex is frequently adulterated by Periplocae Cortex, known as “xiangjiapi”, which originates from the root bark of *Periploca sepium* Bge., belonging to a different family (Asclepiadaceae). Both herbs share similar morphological characteristics, but their functions and efficacies are not quite the same. Periplocae Cortex is a common Chinese herbal medicine with outstanding efficacy in removing edema, expelling wind-dampness and strengthening the bones and muscles. Nowadays, it is mainly used for relieving rheumatic conditions and slaking dropsy, and for treating cardiovascular diseases [6]. Periplocin is a cardiac glycoside compound that has been implicated in various clinical accidents [7]. As is well known, periplocin is not only a main active component but also a potential toxic compound in *P. sepium* Bge.

According to the Chinese Pharmacopoeia, there are approximately 19 kinds of cortex herbs that are commonly used as medicine. These cortex herbs include Acanthopanacis Cortex (Wujiapi) and Periplocae Cortex (Xiangjiapi), and all of them have important medicinal value. Although these medicinal materials are few, distinguishing them from each other can still be confusing because of misidentification or shortage of medicinal sources. Based on market investigation, Acanthopanacis Cortex (Wujiapi) was often confused with Periplocae Cortex (Xiangjiapi). Periplocae Cortex is a common adulterant of Acanthopanacis Cortex in medicine markets and drug stores. Zhao et al. [5] detected five adulterants by using the ITS2 barcode from nine Acanthopanacis Cortex samples purchased from drug stores and medicine markets. In their study, four out of the five adulterants were derived from *P. sepium*, also known as “Periplocae Cortex”. Furthermore, among the unqualified Chinese medicinal materials and TCM decoction pieces in China in 2021, Acanthopanacis Cortex occupied the top 10 unqualified varieties of Chinese medicinal materials. The identification of cortex herbs is currently performed using traditional methods, such as character or microstructure observation. However, these identification methods heavily rely on a high professional level of skill on the part of researchers or users. Identifying cortex herbs with traditional identification methods is difficult, and the wide application of alternatives may influence the efficacy of clinical medication. To avoid the occurrence of periplocin side effects and even potential clinical accidents, it is essential to identify Acanthopanacis Cortex accurately. Since ancient times, CHMs have been identified by their morphological characteristics, odor and taste using sensory analysis, but these methods depend primarily on specific human expertise. Moreover, this process is subjective, requires long-term training, and can be easily affected by external parameters. To standardize the trade of cortex herbs, ensure the stability and reliability of their quality, and guarantee the safety and validity of clinical medication, a simple and accurate method for their identification must be developed. With the development of the large-scale production and circulation of TCM decoction pieces, it has become an urgent problem to establish an online rapid detection and identification technology for the quality and type of TCM decoction pieces. Therefore, there is an urgent need for rapid and simple identification procedures for the rapid inspection of raw herbal materials.

The E-nose is an analytical instrument which basically consists of a combination of an array of chemical sensors and pattern recognition software [8,9,10]. The E-nose is capable of recognizing simple or complex mixtures of organic vapors after an appropriate training period [9,11]. The E-nose, which mimics the human sense of smell to a large degree, focuses on volatile compounds using a variety of sensors producing signals to differentiate chemicals. Nowadays, E-nose technology have been successfully applied in different fields, such as quality assessment of food products [12] and in environmental monitoring [13]. Over the recent years, the E-nose has been increasingly applied to the analysis of CHMs due to their unique smells [14]. The E-nose technology has gradually been adopted to assess the quality of Chinese medicines, and it can be applied to the discrimination of origin [15], authenticity [16], and harvesting time [17]. Compared to the traditional methods, the main advantage of the E-nose is that data normalization can perform odor assessment on a continuous basis with the characteristics of being non-invasive, fast, sensitive and requiring no pretreatment. Examples of its use include a geoherbalism evaluation of Radix Angelica sinensis based on E-nose [18], the quality control of Alpinia officinarum using an E-nose and GC-MS coupled with chemometrics [19], the discrimination and characterization of licorice (*Glycyrrhiza glabra* L.) roots utilizing E-nose and HS-SPME/GC/MS analysis [20], as well as a quality control method for musk using an E-nose coupled with chemometrics [16]. However, there are few reports on odor analysis of Acanthopanacis Cortex and Periplocae Cortex. In recent studies, Gao et al. [21] showed that LC-MS/MS and GC-MS assays, combined with multivariate statistical analysis for Cortex Periplocae, provided a comprehensive and effective means for its quality evaluation. Li et al. [22] selected 23 ingredients as potential Q-markers for Periplocae Cortex based on plant metabolomics and network pharmacology, and distinguished all collected Acanthopanacis Cortex and Periplocae Cortex samples according to an improved PLS-DA model. Among the few studies reporting on the application of E-nose recognition in the research field of Acanthopanacis Cortex and Periplocae Cortex odor, however, there is no relevant report on the material basis of the E-nose response for Acanthopanacis Cortex and Periplocae Cortex. Therefore, E-nose technology can overcome some deficiencies in the current research of Acanthopanacis Cortex and Periplocae Cortex aroma.

Instrumental analysis methods such as gas chromatography–mass spectrometry (GC-MS) can only give the composition and content of the aroma components, but cannot determine the main aroma compounds that play a key role in the aroma. The E-nose can determine which type of characteristic gases the main volatile substances belong to in the sample analysis process through the measurement of aroma compounds. Then, principal component analysis (PCA), linear discriminant analysis (LDA) and sensor differential contribution analysis (Loadings) can be used to effectively determine which type the unknown samples belong to, and thus achieve an experimental result of verifying the unknown samples with the E-nose. Metal oxide semiconductor (MOS) gas sensors, which have the advantages of cross-sensitivity, broad spectrum response and low-cost, have been widely used in E-nose applications. MOS-based gas sensors have been studied for many years, and several commercially available E-noses based on this technology are now available, such as PEN3 from Airsense Analytics and Fox 4000 from Alph Mos [9,14]. In this study, an E-nose (PEN3) equipped with an array of MOS sensors was employed to analyze the two cortex herbs. The response values of the E-nose were recorded and analyzed by PCA, which made possible the extraction of information based on the overall properties of the sample and thus perform a classification without the need for additional compositional data.

Compared with traditional odor analysis methods such as GC-MS and Fourier Transform infrared spectroscopy (FT-IR), an E-nose is an easy system to build. It has the characteristics of needing only simple sample pretreatment, being non-destructive, providing relatively fast evaluation and detection, and having a wide odor detection range. It usually has high sensitivity and selectivity for the detected odor. Furthermore, the information obtained by the different sensors of the E-nose represents the overall distribution of all volatiles in the sample, rather than the amount of specific components or components that would normally be measured analytically. If E-nose technology is comprehensively applied with GC-MS, it can make up for the ambiguity, subjectivity and inaccuracy of human sensory description. Thus, in this study, the E-nose detection technology was used to analyze the volatile components of Acanthopanacis Cortex and Periplocae Cortex, and the volatile components were further determined by GC-MS fingerprint. Combined with multivariate statistical analysis, the differences of volatile components were further discussed, and the potential markers with the greatest contribution to the differences were explored. Analysis of variance (ANOVA), PCA and supervised orthogonal partial least squared discriminant analysis (OPLS-DA) were applied to process and analyze the experimental data. In this study, E-nose, GC-MS and chemometrics methods were used to differentiate Acanthopanacis Cortex and Periplocae Cortex by their volatile constituents. The E-nose was first introduced to differentiate Acanthopanacis Cortex and Periplocae Cortex rapidly and objectively. It provides a more scientific basis for odor recognition, and it lays the foundation for the discussion and development of the specificity and exclusivity of the array of E-nose sensors in CHMs, in order to provide a reference for the establishment of rapid detection and identification technology of TCM decoction pieces and the online detection and identification of large-scale production and circulation in the future.

## 2. Results

### 2.1. Differentiation of Acanthopanacis Cortex and Periplocae Cortex by E-Nose

#### 2.1.1. PCA

Experimental samples were analyzed using unsupervised PCA. The response values of the sensors were analyzed by PCA, and the total variance does not change with the mathematical transformation. The first variable, which has the largest variance, is known as the first principal component. The second variable, irrelevant to the first variable, is called the second principal. Figure 1 shows the PCA score plot and loading plot that provide an indication of the variables’ contributions in the discrimination of different herb species. As observed in Figure 1A, the first two principal components led to a total variance of 83.5%, of which 54.5% was explained by PC1 and 29.0% was explained by PC2. As shown in the PCA loading plot (Figure 1B), Periplocae Cortex formed a cluster at the left side of the biplot, while the remaining Acanthopanacis Cortex samples clustered at the right side. Moreover, most of the E-nose sensors clustered to the left, including W_1_W, W_1_S, W_2_S and W_3_S, reflecting associations between these sensors and the Periplocae Cortex samples that also clustered at the left side of the biplot. In contrast, Acanthopanacis Cortex samples that clustered at the right side of the biplot showed a stronger association with W_5_S, W_6_S, W_2_W, W_3_C, W_1_C and W_5_C.

The sensors were capable of differentiating Acanthopanacis Cortex and Periplocae Cortex, even though the geographic origins and production years were different in each species. However, the samples from different geographic origins or different production years could not be separated within each group, which implied that there was no significant difference of volatile constituents among samples with different geographic origins or production years within the same species. The portable E-nose was able to distinguish Acanthopanacis Cortex in combination with PCA. The determination was rapid, and only a small amount of the sample was required. The overall procedure was eco-friendly because no solvent was required in the process. The portability of this device also makes it promising for rapid on-site analysis.

#### 2.1.2. OPLS-DA

As an unsupervised analysis method, PCA analysis only reflects the original state of the data and observes the natural distribution and group relationship of the test samples, but it cannot ignore intra-group errors, eliminate random errors irrelevant to the research purpose, or ignore the overall characteristics and change rules of the data, which is not conducive to finding the differences between groups and the metabolites of differences. In order to determine the chemical differences between Acanthopanacis Cortex and Periplocae Cortex, the data were further analyzed by supervised OPLS-DA analysis. OPLS-DA was used to assess and maximize the differences between the two groups. The OPLS-DA score plot and loading plot are shown in Figure 2. An obvious separation trend between the two groups was observed in the OPLS-DA score plot (Figure 2A), which proved that the model was successfully established. At the same time, the model parameters R^2^X = 0.804, R^2^Y = 1, Q^2^ = 0.844 in the model indicated that the model had a good fitness and prediction. As shown in the OPLS-DA loading plot (Figure 2B), the Periplocae Cortex samples that clustered at the right side of the biplot showed a stronger association with W_1_S, W_2_S and W_3_S.

#### 2.1.3. Relationship between Sample Odor and E-Nose Sensor

Redundancy analysis (RDA), as a constrained multivariate statistical method, is mainly used to explain the variability of response variables as much as possible through the linear combination of explanatory variables. Redundancy analysis was used to visually compare the correlation of the test results with the two-way data of the sensor and volatile compounds in this study, and the results are shown in Figure 3. The results show that the cumulative contribution of PCA principal components is 83.5%, indicating that it retains most of the information in the original data. The difference among samples is obvious in PCA differentiation, and the difference is mainly reflected in the vertical axis with information weight of PC2 (29.0%). At the same time, it can be seen that sensors W_5_C/W_1_C/W_3_C/W_2_W tend to Acanthopanacis Cortex, while W_2_S/W_1_S/W_3_S tend to Periplocae Cortex. Considering the characteristics of the sensors (Appendix A), W_5_C/W_1_C/W_3_C/W_2_W are more sensitive to aromatic components such as benzene, ammonia, short-chain alkane aromatic components and organic sulfur compounds, while W_2_S/W_1_S/W_3_S are more sensitive to methane, alcohols, aldehydes and ketones and long-chain alkanes. Therefore, it can be inferred that the aromatic components benzene, ammonia, short-chain alkane aromatic components, organic sulfur compounds, methane, alcohols, aldehydes and ketones and long-chain alkanes have important relations with the classification of medicinal materials.

### 2.2. Composition and Relative Contents of Volatile Aroma Compounds in Acanthopanacis Cortex and Periplocae Cortex

A total of 82 volatile compounds were tentatively identified based on the mass spectra using the National Institute of Standards and Technology (NIST) 17 L Mass Spectra Database, as well as comparison with the literature. Table 1 shows the composition details of the volatile oils of Acanthopanacis Cortex and Periplocae Cortex.

### 2.3. Metabonomics Difference Analysis of Volatile Aroma Compounds in Acanthopanacis Cortex and Periplocae Cortex

#### 2.3.1. Chemometric Analysis

PCA is usually used as a first step in chemometric analysis to visualize grouping trends and outliers. Principle component 1 versus principle component 2 scores plots of test samples are shown in Figure 4. Without using class information, Acanthopanacis Cortex and Periplocae Cortex were clearly separated in the PCA scores plot.

To find chemical markers that were responsible for such separation, OPLS-DA was performed. In the OPLS-DA model, class separation was found in the first predictive component, also referred to as the correlated variation, and variation not related to class separation was seen in orthogonal components (Figure 5A). The model quality was described by the goodness-of-fit parameter R^2^, which represents the total explained variation for the X matrix, and the predictive ability parameter Q^2^. The separation of predictive and orthogonal components facilitates model interpretation.

In the OPLS-DA model, R^2^X and R^2^Y represent the interpretation rate of the model for the X and Y matrices, respectively, and Q^2^ represents the prediction ability of the model. Theoretically, the closer R^2^ and Q^2^ are to 1, the better the model is, and the lower R^2^ and Q^2^ are, the worse the fitting accuracy of the model is. In general, R^2^ and Q^2^ higher than 0.5 (50%) is better, and higher than 0.4 is acceptable, and the difference between the two should not be too large. It can be seen from Figure 5A that R^2^X = 0.428, R^2^ = 0.987 and Q^2^ = 0.957 in the model, where R^2^X = 0.428 indicates that the model can reflect 42.8 % of the data changes. R^2^ and Q^2^ are close to 1.0, indicating that the model has good explainability and fitting degree. The two groups of samples had good clustering on the OPLS-DA dispersion point map, the differences within the groups were small, and the samples were completely separated between different groups. In order to avoid the overfitting phenomenon where the OPLS-DA model can effectively distinguish inter-group samples but cannot effectively predict new sample data sets, permutation test and cross-validation analysis (CV-ANOVA) in SIMCA 14.1 were used to verify the reliability of the model. The results of the replacement test are shown in Figure 5A. The abscissa in the figure represents the retention of the sample during the permutation test, and the point at which the retention equals 1.0 is R^2^ and Q^2^ obtained by the original OPLS-DA model. In the process of permutation test, if all R^2^ and Q^2^ are lower than the value of permutation reservation equal to 1.0, and the regression line at Q^2^ crosses the abscissa or is less than 0, the intercept is generally considered to be negative, and the statistical model is valid without over-fitting [23]. It can be seen from Figure 5B that after 200 times of cross-validation, the regression line of model Q^2^ still intersects with the abscissa, and the intercept intersecting with the ordinate is less than 0, indicating that the model has not been over-fitted. At the same time, the significance probability value of the cross-validation analysis results was *p* = 5.08738 × 10^−16^ < 0.05, indicating that the OPLS-DA model established in this study was stable and reliable, with statistical significance.

#### 2.3.2. Comparative Analysis of Potential Chemical Markers

Multivariate variable importance in projection (VIP) values calculated in the OPLS-DA model were used to screen variables that contribute to class separation. VIP is the weight value of an OPLS-DA model variable, which can be used to measure the influence intensity and explanatory ability of accumulation difference of each component on classification and discrimination of each group of samples. The larger the VIP value is, the greater the contribution rate is, and VIP > 1 is a common screening criterion for differential metabolites [24]. As can be seen from Figure 5C, there were 24 compounds with VIP > 1. In order to make the analysis results more accurate, a nonparametric test (Mann–Whitney U test) was used to analyze the compounds with VIP > 1 [25], and the analysis results are shown in Appendix A. The probability value of 24 compounds was <0.05, and thus significant. In conclusion, there are 24 different markers among the the two kinds of herbs.

In total, 24 volatile components were screened out. The representative volatile compounds of Periplocae Cortex were mainly 2-Hydroxy-4-methoxybenzaldehyde (No. 33), methyl 2-hydroxy-4-methoxybenzoate (No. 68), 5-formyl-2-methoxyphenyl acetate (No. 66), 3,4,5-trimethyloxolan-2-one (No. 62), Dodecane (No. 17), 2,4-ditert-butylphenol (No. 71) and 3-Hydroxy-4-methoxybenzaldehyde (No. 64). The representative volatile compounds of Acanthopanacis Cortex were mainly 2-Hydroxy-4-methoxybenzaldehyde (No. 33), (1R,5S)-6,6-dimethyl-bicyclo[3.3.1]hept-2-en-2-carbaldehyde (No. 16), (1S,5S)-7,7-dimethyl-4-methylidenebicyclo[3.1.1]heptan-3-ol (No. 8), (1R,5S)-2-methyl-5-prop-1-en-2-ylcyclohex-2-en-1-ol (No. 20), 2,7,7-trimethylbicyclo[3.1.1]hept-2-en-4-one (No. 25), 2-(4-methyl-1-cyclohexa-2,4-dienyl)propan-2-ol (No. 12), 1-(2,2,3-trimethyl-1-cyclopent-3-enyl)ethanone (No. 7), 2-[(1R)-4-methyl-1-cyclohex-3-enyl]propan-2-ol (No. 15), (1S,5S)-2,7,7-trimethylbicyclo[3.1.1]hept-2-en-4-one (No. 19), 2-(4-methylidene-1-cyclohex-2-enyl)propan-2-ol (No. 10), (5S)-2-methyl-5-prop-1-en-2-ylcyclohex-2-en-1-one (No. 24), 7,7-dimethyl-4-methylidenebicyclo[3.1.1]heptan-3-one (No. 11), (1R)-4-methyl-1-propan-2-ylcyclohex-3-en-1-ol (No. 13), 2-(4-methylphenyl)propan-2-ol (No.14), Dodecane (No. 17), (1aR,4aR,7S,7aR,7bR)-1,1,7-Trimethyl-4-methylenedecahydro-1H-cyclopropa[e]azulen-7-ol (No. 51), 5-(5,5,8a-trimethyl-2-methylidene-3,4,4a,6,7,8-hexahydro-1H-naphthalen-1-yl)-3-methylpent-1-en-3-ol (No. 58), (1R,4R,6R,10S)-4,12,12-Trimethyl-9-methylene-5-oxatricyclo[8.2.0.0]dodecane (No. 52) and [(4S)-4-prop-1-en-2-yl-1-cyclohexenyl]methanol (No. 29). This result indicated that the compounds were probably responsible for the observed separation (VIP > 1, *p* < 0.05) (Appendix A), constituting 77.71% and 97.12% of the total content in Acanthopanacis Cortex and Periplocae Cortex, respectively. 2-Hydroxy-4-methoxybenzaldehyde (No. 33) and Dodecane (No. 17) are found in both herbs, with higher levels in Periplocae Cortex (91.74% and 1.54%, respectively).

## 3. Discussion

### 3.1. Acanthopanacis Cortex and Periplocae Cortex Should Be Correctly Identifed and Used

Acanthopanacis Cortex and Periplocae Cortex are similar in morphological characteristics and difficult to distinguish from each other. Acanthopanacis Cortex and Periplocae Cortex always use the names “Nanwujiapi” and “Beiwujiapi” in medicine markets, respectively. These two names sound similar and can be easily confused. Acanthopanacis Cortex and Periplocae Cortex belong to different families and genera, and they have different effects. Although they both have the effect of removing wind and dampness, they should be used in strictly different ways due to their different sources and components, and they cannot be substituted for each other in clinical practice. If it is to dispel wind dampness and strengthen the liver and kidneys, Acanthopanacis Cortex should be used. If it is to dispel wind dampness, strengthen the heart and promote water, Periplocae Cortex should be used.

Acanthopanacis Cortex is increasingly becoming popular in clinical applications. However, several closely related species of E. nodiflorus are locally used as Acanthopanacis Cortex in several places [26,27]. Acanthopanacis Cortex is easily confused with other herbs in medicine markets, thereby causing potential safety issues. Periplocae Cortex is a common adulterant of Acanthopanacis Cortex in medicine markets and drug stores. Periplocae Cortex extract contains periplocin and it is well-known that periplocin is poisonous and has a cardiotoxic effect which is similar to that of digitalis. The adverse reaction of Periplocae Cortex may be caused by the large difference in the content of glucoside in the decoction pieces of Periplocae Cortex, the different degree of freshness of medicinal materials, the different extraction rate of glucoside caused by different medication methods, the misuse of Periplocae Cortex as Acanthopanacis Cortex and the unreasonable combination of medication [28]. Acanthopanax gracilistylus wine poisoning has been reportedly caused by Periplocae Cortex substitution for Acanthopanacis Cortex. Moreover, Acanthopanacis Cortex is seriously mistaken for Periplocae Cortex when used in Chinese patent medicines or health supplement products [29].

Existing toxic adulterants are important factors causing safety issues. In addition, adulterants mixed with non-medicinal parts are still severe problems and challenges in the current quality of CHMs [22]. However, identifying cortex herbs using traditional identification methods is difficult, especially when the cortex is dried and sliced [30,31]. Thus, correct identification between Acanthopanacis Cortex and Periplocae Cortex is absolutely essential to ensure clinical safety.

### 3.2. E-Nose Effectively Identifies Acanthopanacis Cortex and Periplocae Cortex

Traditional identification methods cannot easily authenticate sliced, shredded, or simply processed herbal medicine. The morphological authentication approach largely depends on taxonomists and becomes infeasible because of the absence of identifying features. The E-nose offers the potential to resolve this problem. In the present research, the studied materials mainly included cortexes from medicine markets and drug stores. The four samples numbered X-10, X-11, X-12 and X-13 were from samples for supervision and inspection, and the sample name was originally labeled as Acanthopanacis Cortex. According to the character and microscopic identification, they were actually samples of Periplocae Cortex which were misused as Acanthopanacis Cortex. Our research demonstrated that the E-nose can also distinguish these four samples from other samples. Thus, the E-nose is a good tool to identify Acanthopanacis Cortex and Periplocae Cortex accurately.

### 3.3. GC-MS Combined with Multivariate Statistical Analysis Effectively Classifies and Identifies Acanthopanacis Cortex and Periplocae Cortex

For the varieties with a high content of volatile oil and the essential oil as the main effective component of its efficacy, the method of steam distillation was used to extract the essential oil and GC-MS was used to identify the difference of the essential oil components of the crude drugs from different origins. This method is simple, rapid and specific, and can be used for the quality control of CHMs effectively. In this study, the unknown components of Acanthopanacis Cortex and Periplocae Cortex were tentatively identified by total scanning mass spectrometry. In the preparation of Periplocae Cortex samples, n-hexane ultrasonic extraction was used to extract volatile oil, but steam distillation was not used to extract volatile oil, which reduced the dosage of medicinal materials and the tedious oil extraction process, and the method was simple and timesaving. It was proved that n-hexane ultrasonic extraction detected relatively more components in Periplocae Cortex. In this study, GC-MS technology and multivariate statistical analysis were used to analyze the differences of volatile chemical components of Acanthopanacis Cortex and Periplocae Cortex, and a total of 82 components, including nine co-contained components, were extracted by chromatographic peak integration and matching. PCA and OPLS-DA methods were used for data processing to analyze the chemical composition differences among different samples (Figure 2). In the scores plot for OPLS-DA, the model parameters (R^2^Y = 1 and Q^2^ = 0.957) show that it has a high explained variance (R^2^Y) and cross-validated predictive capability (Q^2^), and the results of cross-validation and the low intercepts (R^2^ = 0.327 and Q^2^ = −0.365) show that there was no over-fitting in the model. In total, 24 compounds were found with the VIP value larger than 1.0. Our research demonstrated that the method based on GC-MS combined with multivariate statistical analysis could effectively classify and distinguish two species from one another, and both unsupervised PCA analysis and supervised OPLS-DA analysis can distinguish these two kinds of cortex herbs well.

### 3.4. The Standards for Periplocae Cortex Need to Be Further Improved

In the 2020 edition of the Chinese Pharmacopoeia, only 4-methoxylsalicylicaldehyde (the same as compound 33, namely, “Benzaldehyde,2-hydroxy-4-methoxy”) was used for a content determination item of Periplocae Cortex [32], but this ingredient does not represent the efficacy and toxicity of Periplocae Cortex, so it cannot effectively control the quality of Periplocae Cortex. It has been reported that an HPLC method has been established for the simultaneous determination of periplocin and 4-methoxysalicylicaldehyde in Periplocae Cortex, which can better control the quality of the medicinal materials [28,33,34]. It is suggested that the content determination of periplocin should be increased to control the quality of Periplocae Cortex strictly.

### 3.5. E-Nose Technology Is Expected to Become a Technical Tool for Quality Supervision and Improvement of CHM_S_

Odor characteristics are one of the evaluation indexes reflecting the intrinsic quality of CHM_S_. The overall odor characteristics of CHM_S_ are directly related to the types and contents of the chemical components contained in them, which is the correlation point between the external attributes and intrinsic quality of CHM_S_. The traditional fingerprint lays more emphasis on the characterization of internal components, while the odor fingerprint lays more emphasis on the expression of volatile component signals of CHM_S_. For the CHM_S_ with high correlation between odor and quality, or obvious odor characteristics but not strong fingerprint characteristics, it can reflect the overall odor-quality characteristics of the CHM_S_, and objectively express the “smell of other taste” in traditional identification experience. The objective evaluation of odor by an E-nose makes up for the shortcomings of traditional identification methods such as fuzziness, subjectivity and inaccuracy, the limitation of one or a class of chemical components as indicators and the complexity of the fingerprint method. The smell of CHM_S_ is related to their chemical composition. The application of E-nose technology in quality control for CHM_S_ is still at the laboratory research stage, but there is no report on the application of E-nose technology in sampling inspection and supervision and inspection of CHM_S_. The E-nose technology is here used to establish the fingerprint of CHM odor in the sampling inspection and supervision and inspection of CHM_S_. The quantitative expression of “odor” is used to supplement sampling inspection and improve supervision, which can identify fake and inferior CHM decoction pieces and enhance “targeted supervision and targeted sampling inspection”. The odor fingerprint of CHM_S_ was established by E-nose to achieve the objective data expression of odor and make odor become a quantifiable index. The E-nose combined with GC-MS analysis interpreted the inner relation of variety and quality; at the same time, in combination with the pattern recognition method, the smell can be used to completely control the quality of CHM_S_, as new quantitative indicators, and provide new opportunities for further research into CHM_S_; thus, the E-nose combined with GC-MS and multivariate statistical analysis is convenient for medicine market supervision. Odor can be a new quantitative index to control the quality of CHM_S_, providing a new opportunity for the in-depth study of CHM_S_, so as to improve the overall level of the CHM industry and promote the modernization of CHM_S_. E-nose technology has become a new technology to distinguish the authenticity of CHM_S_. E-nose technology is expected to become a technical tool for quality supervision and improvement of CHM_S._

The gas sensor is the most critical component in the E-nose system. MOS sensor technology has become relatively mature. Many studies [35,36] have reported that surface modification can be used to improve the performance of the sensor. At present, most gas sensors still have shortcomings in selectivity, stability and applicability. Many researchers have begun to try to combine the E-nose with mature analysis methods, such as gas chromatography and mass spectrometry [37], to obtain more odor information data, improve the selectivity of the E-nose and the dimension and diversity of response patterns, and enhance the recognition ability of the E-nose for complex mixed gases. In addition, investigation of appropriate pattern recognition methods also can provide satisfactory recognition accuracy [38,39]. In this study, the number of gas sensors in the E-nose system was limited and each gas sensor was usually sensitive to one kind of odor; it is crucial to identify more sensitive, selective and stable sensing materials to construct the sensor arrays. Therefore, the challenge for E-nose technology is to investigate new materials and make it more portable and more sensitive with faster response times when exposed to different volatile species.

## 4. Materials and Methods

### 4.1. Plant Materials

A total of 29 samples belonging to two species, including Acanthopanacis Cortex (16 cortex samples) and Periplocae Cortex (13 cortex samples), were collected from medicine markets, drug stores, manufacturers and hospitals. All samples were identified by the laboratory; detailed information about the samples is shown in Appendix A. The samples were stored in airtight containers in a cool (10–20 °C) dark room.

### 4.2. Reagents

N-hexane (Lot: 20190128) was purchased from Shanghai Lingfeng Chemical Reagent Co., Ltd. (Shanghai, China), and Anhydrous sodium sulfate (Lot: 20210406) from Sinopharm Chemical Reagent Co., Ltd. (Shanghai, China).

### 4.3. E-Nose Analysis

Analyses were conducted with a portable E-nose device (PEN3, Airsense Analytics GmbH, Schwerin, Germany). The device was composed of a sampling apparatus, a sensor chamber containing an array of 10 metal oxide semiconductor type chemical sensors (namely W_1_C, W_5_S, W_3_C, W_6_S, W_5_C, W_1_S, W_1_W, W_2_S, W_2_W and W_3_S), and pattern recognition software for data recording and analysis. The detection limit of the hot sensors was in the range of 1 ppm. The selectivity of the sensors was determined by the sensing material, the dopant material, the working temperature and the geometry of the sensor. Sensors with good selectivity for sulfur organic compounds, methane, hydrogen, alcohol and hydrocarbons were used. The used sensors and their main attributes are described in Appendix A [40].

The analytical system has a special sampling system integrated which, by means of an automatic control (autoranging), prevents an overloading of the sensors and also leads to better and faster qualitative and quantitative results. The E-nose is able to detect complex mixtures of gaseous compounds. Smells can be learned and recognized. In the process, the E-nose immediately notices deviations from the “standard smell” it has learned to identify. The response values of the E-nose sensors were expressed as the ratios of conductance (G/G_0_), where G and G_0_ were the conductivities of the sensors when the sample gas and reference air flowed over the measurement chamber, respectively. In this study, the stable response values of each sensor were used for later analysis; Acanthopanacis Cortex and Periplocae Cortex were at 90 s. The results were determined through statistical methods such as euclid, correlation, factor analysis (PCA) or discriminant function analysis (DFA).

The ground samples (samples were passed through an 80 mesh sieve) were weighed to 3.0 g and then placed in 20 mL headspace vials, sealed with a silicone/PTFE septum and magnetic caps, and then stored at 25 °C for 10 min until analysis. During the measurement process, a needle connected to Teflon tubing was used to penetrate the septum and the headspace gas was pumped into the sensor chamber at a constant rate of 300 mL/min. The measurement phase lasted 90 s, and the response values of the E-nose were recorded by a computer every second. When the measurement was finished, the cleaning phase was activated, which lasted 100 s. The main purpose was to clean the test chamber and return the sensors to their baseline values. All analyses were performed in triplicate.

### 4.4. Extract of Volatile Oil and GC-MS Analysis

#### 4.4.1. Extraction of Volatile Oil

Steam distillation, a typical extraction method for volatile oils, was chosen according to the Chinese Pharmacopoeia [32]. The dried powder (20 g) of Acanthopanacis Cortex was accurately weighed and transferred to a 500 mL round-bottomed flask soaked in 500 mL of water. Water was added from the top of the volatile oil determination apparatus until the water spilled onto the round-bottomed flask. Then, the essential oils were extracted by water distillation for 6 h. Volatile oil was separated from the water layer and leached into the n-hexane layer, diluted the n-hexane to 25 mL, and then the n-hexane layer was dried over anhydrous sodium sulfate. The samples were stored at 4 °C in the refrigerator before GC-MS/MS analysis. The dried powder (1.0 g) of Periplocae Cortex was taken and placed in a tapered bottle with a plug. A total of 15 mL of n-hexane was added to the powder, which was packed and weighed. Ultrasonic treatment (power 400 W, frequency 50 kHz) was performed for 30 min, cooling was performed, and it was weighed again. Then, the n-hexane layer was dried over anhydrous sodium sulfate. The samples were stored at 4 °C in the refrigerator before GC-MS/MS analysis.

#### 4.4.2. Instrumentation and GC-MS Conditions

The GC-MS analyses were performed using gas chromatography coupled to tandem mass spectrometry (GC-MS) on an Agilent 8890/7000D Triple Quadrupole (Agilent 8890/7000D, Santa Clara, CA, USA). Chromatographic separations were conducted on an HP-5MS (30 m × 0.25 mm, 0.25 μm film thickness) capillary column (Agilent19091S-433, Santa Clara, CA, USA). For the analysis of Acanthopanacis Cortex samples, the oven temperature was initially programmed at 60 °C, 10 °C/min to 85 °C, 1 °C/min to 95 °C, and finally 10 °C/min to 180 °C and holding for 10 min. For the analysis of Periplocae Cortex samples, the oven temperature was initially programmed at 60 °C, 8 °C/min to 200 °C, and holding for 5 min. High-purity (99.999%) helium was used as the carrier gas at a flow rate of 0.8 mL/min. The injection method was splitless injection, the injection volume was 1 μL and the injection temperature was 250 °C. The spectrometers were operated in the electron-impact (EI) mode and full-scan mode (*m*/*z* 35–550), the ionization energy was 70 eV, and the electron multiplier was 1204.3 V. The temperatures of the injection port, ionization source and transfer line were set at 250 °C, 230 °C and 250 °C, respectively.

### 4.5. Data Analysis

For the E-nose analysis, the stable value of each sensor was selected as the characteristic value, and the mean values of the data obtained from the E-nose after repeated experiments were processed using the statistical software SPSS26.0 (SPSS Inc., Chicago, IL, USA). All imported data were mean centered for the multivariate analysis. PCA is an unsupervised method that reduces multidimensional data into orthogonal coordinates based on maximum variance by linear projection. By employing PCA, data are transformed into two-dimensional (2D) or three-dimensional (3D) coordinates. In the 2D and 3D PCA plots, samples with similar patterns are located together and the differences between groups can be visualized [41]. In this study, PCA was used to derive the first two principal components from the E-nose data and to visualize the information present in the data. PCA was used to provide an overview for all of the groups and OPLS-DA was utilized to maximize the discrimination and present the differences in the volatile organic compounds between all of the groups. Biplot analysis was used to visually compare the correlation of the test results with the two-way data of the sensor and volatile compounds in this study.

For the GC-MS analysis, the volatile constituents were tentatively identified by comparing the mass spectra with the National Institute of Standards and Technology (NIST) 17 L Mass Spectra Database, as well as comparison with the literature [6,42,43,44,45,46,47,48,49]. The relative contents of volatile organic compounds (VOCs) were calculated using the area normalization method. Simca P 14.1 (Umetrics, Umea, Sweden) and R software (version 4.0.3, https://www.r-project.org/) were used for plotting, data processing and principal component analysis. PCA, a supervised pattern recognition method, was first employed to visualize the global variance of the data sets and find outliers. To maximize the separation between samples, the OPLS-DA model was applied to maximize covariance between the measured data (X variable, relative content) and the response variable (Y variable, predictive classifications), and simultaneously to remove non-correlated variation in X variables to Y variables or variability in X that is orthogonal to Y [50,51]. Hotelling’s T2 region, shown as an ellipse in the scores plot, defines the 95% confidence interval of the modeled variation. The quality of the models was described by R^2^ and Q^2^ values. R^2^ is defined as the proportion of variance in the data explained by the models and indicates goodness of fit, and Q^2^ is defined as the proportion of variance in the data predictable by the model, and indicates predictability [52]. In addition, to validate the model, permutation tests were performed where the Y variable was permuted randomly 200 times and OPLS-DA models were created between the metabolites data and the permutated Y variables, highlighting metabolites having stronger correlation to the original Y variables compared to permutated Y variables. The variable importance in projection (VIP) value from the orthogonal PLS-DA (OPLS-DA) model combined with Mann–Whitney U test were used to search the differential volatile components (VIP > 1 and *p* < 0.05).

Unsupervised PCA and supervised OPLS-DA were performed on the data using Simca P 14.1 software (Umetrics, Umea, Sweden). Simca P 14.1 and R software were used for plotting, data processing and principal component analysis. Significance in differences of metabolites between groups were evaluated for individual values using the nonparametric test (Mann–Whitney U test) in SPSS.

## 5. Conclusions

In this study, an E-nose, GC-MS and multivariate statistical analysis were first applied to differentiate between Acanthopanacis Cortex and Periplocae Cortex. The differentiation of Acanthopanacis Cortex and Periplocae Cortex was successfully achieved without reliance on appearance characteristics. These methods offer valuable techniques to visualize the relationships of Acanthopanacis Cortex and Periplocae Cortex according to their chemical composition. The E-nose provided an objective way to differentiate Acanthopanacis Cortex and Periplocae Cortex by their odors with the advantages of being rapid and easy to use. The GC-MS analysis revealed the differences between the chemical profiles of the volatile constituents of Acanthopanacis Cortex and Periplocae Cortex, and we determined that 24 constituents can be used as chemical markers to distinguish these two species by employing multivariate statistical analysis. The proposed methods are rapid, simple, eco-friendly and can successfully differentiate between Acanthopanacis Cortex and Periplocae Cortex using their odors.

## Figures and Tables

**Figure 1 molecules-27-08964-f001:**
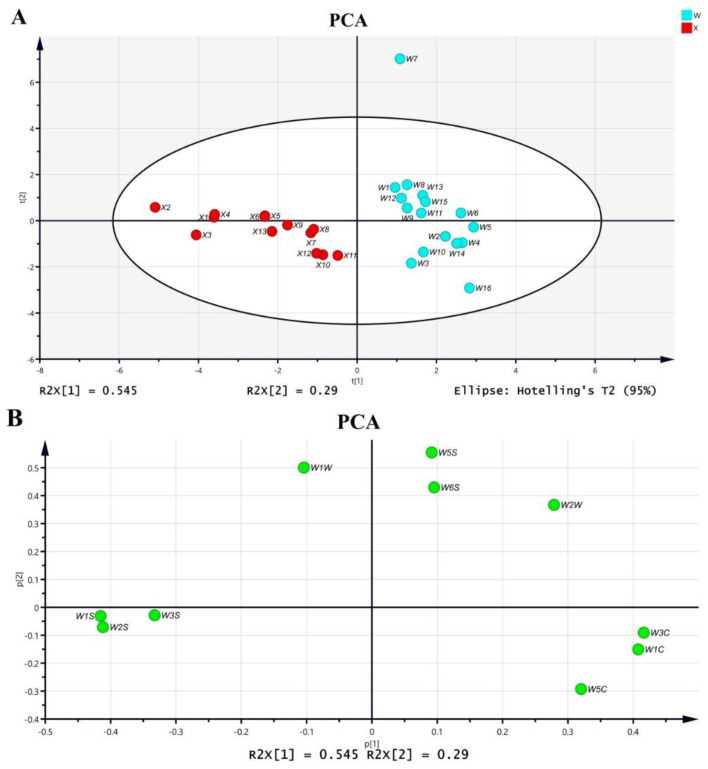
PCA plot of flavor profiles among the herbs. (**A**) PCA score plot representing the samples; (**B**) PCA loading plot representing the variables.

**Figure 2 molecules-27-08964-f002:**
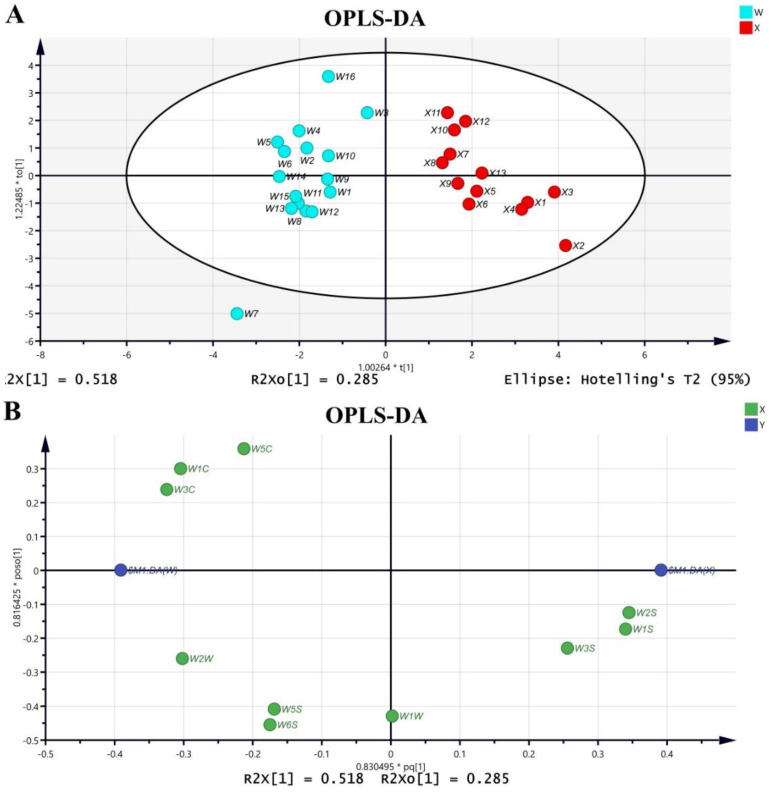
OPLS-DA of Acanthopanacis Cortex and Periplocae Cortex. (**A**) OPLS-DA score plot representing the samples; (**B**) OPLS-DA loading plot representing the variables.

**Figure 3 molecules-27-08964-f003:**
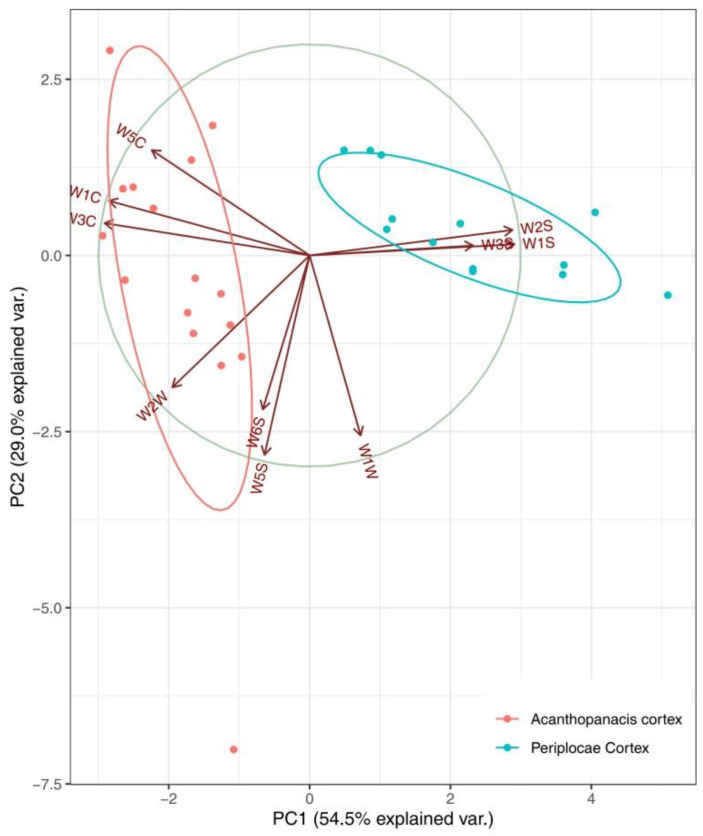
RDA biplot displaying the relationship between sample odor and E-nose sensors.

**Figure 4 molecules-27-08964-f004:**
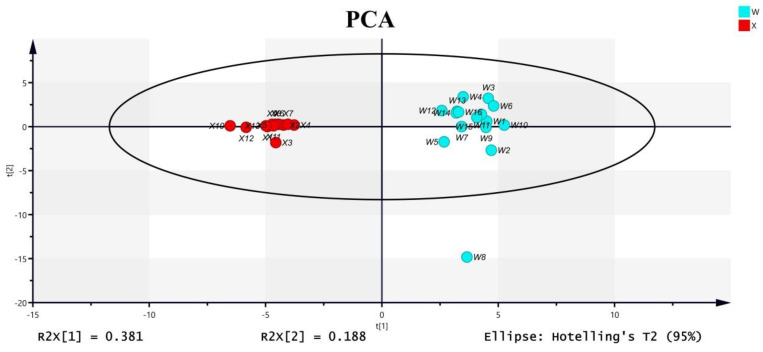
PCA classification results of Acanthopanacis Cortex (W) and Periplocae Cortex (X) based on GC-MS data.

**Figure 5 molecules-27-08964-f005:**
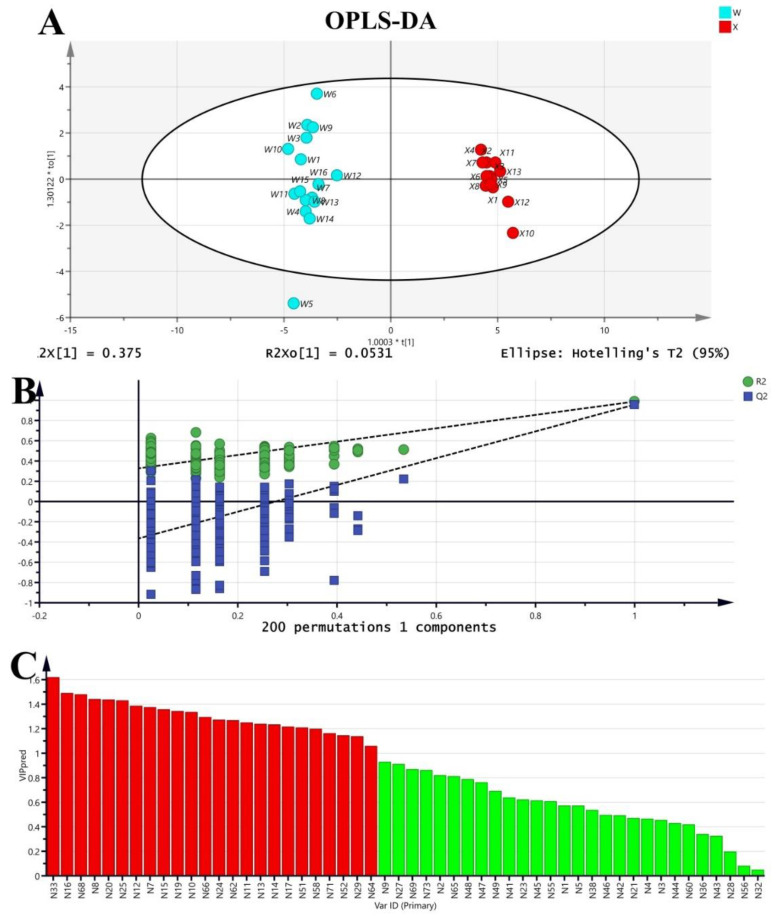
OPLS-DA model of Acanthopanacis Cortex (W) and Periplocae Cortex (X) based on GC-MS data. (**A**) OPLS-DA model; (**B**) 200 permutation tests; (**C**) identification of key VOCs by OPLS-VIP. The horizontal coordinate number was corresponding to Table 1; red indicated volatile aroma compounds with VIP value greater than 1; green indicated volatile aroma compounds with VIP value less than 1.

**Table 1 molecules-27-08964-t001:** The composition and relative contents of VOCs in Acanthopanacis Cortex and Periplocae Cortex (*n* = 3).

No.	Retention Time/Min		Constituents	Formula	Compatibility		Relative Contents (% Average)		Structure Type
	W	X			W	X	W	X	
1	6.195	-	1-methyl-4-prop-1-en-2-ylcyclohexa-1,3-diene	C_10_H_14_	92.64	-	0.72 ± 0.34	-	Monoterpenoids
2	6.731	-	1-methyl-2-propan-2-ylbenzene	C_10_H_14_	93.19	-	1.94 ± 0.6	-	Monoterpenoids
3	6.872	-	(4R)-1-methyl-4-prop-1-en-2-ylcyclohexene	C_10_H_16_	86.85	-	0.25 ± 0.16	-	Monoterpenoids
4	7.81	-	1-methyl-4-propan-2-ylcyclohexa-1,4-diene	C_10_H_16_	92.43	-	0.2 ± 0.12	-	Monoterpenoids
5	8.417	-	1-methoxy-2-propylbenzene	C_10_H_14_O	82.38	-	0.17 ± 0.08	-	Aromatic ethers
6	8.933	-	1-Ethenyl-3,5-dimethylbenzene	C_10_H_12_	80.21	-	0.19 ± 0.19	-	Monoterpenoids
7	10.519	-	1-(2,2,3-trimethyl-1-cyclopent-3-enyl)ethanone	C_10_H_16_O	96.46	-	6.65 ± 0.72	-	Monoterpenoids
8	11.171	-	(1S,5S)-7,7-dimethyl-4-methylidenebicyclo [3.1.1]heptan-3-ol	C_10_H_16_O	95.71	-	10.98 ± 0.97	-	Monoterpenoids
9	11.421	-	(1S,2R,5S)-4,6,6-trimethylbicyclo[3.1.1]hept-3-en-2-ol	C_10_H_16_O	92.19	-	3.82 ± 0.94	-	Monoterpenoids
10	11.63	-	2-(4-methylidene-1-cyclohex-2-enyl)propan-2-ol	C_10_H_16_O	85.82	-	4.46 ± 0.53	-	Alkenes
11	12.39	-	7,7-dimethyl-4-methylidenebicyclo[3.1.1]heptan-3-one	C_10_H_14_O	93.79	-	1.76 ± 0.25	-	Monoterpenoids
12	12.58	-	2-(4-methyl-1-cyclohexa-2,4-dienyl)propan-2-ol	C_10_H_16_O	94.68	-	12.13 ± 1.27	-	Alkanes
13	13.118	-	(1R)-4-methyl-1-propan-2-ylcyclohex-3-en-1-ol	C_10_H_18_O	90.29	-	1.04 ± 0.15	-	Monoterpenoids
14	13.448	-	2-(4-methylphenyl)propan-2-ol	C_10_H_14_O	87.44	-	1.61 ± 0.25	-	Alcohols
15	13.731	-	2-[(1R)-4-methyl-1-cyclohex-3-enyl]propan-2-ol	C_10_H_18_O	93.2	-	1.27 ± 0.14	-	Monoterpenoids
16	13.942	-	(1R,5S)-6,6-dimethyl-bicyclo[3.3.1]hept-2-en-2-carbaldehyde	C_10_H_14_O	95.17	-	9.84 ± 0.72	-	Monoterpenoids
**17**	**14.044**	**9.127**	**Dodecane**	**C_12_H_26_**	**84.64**	**97.3**	**0.12 ± 0.12**	**1.54 ± 0.25**	**Alkanes**
18	14.194	-	1,7,7-trimethylbicyclo[2.2.1]heptan-6-ol	C_10_H_18_O	85.15	-	0.04 ± 0.04	-	Monoterpenoids
19	14.446	-	(1S,5S)-2,7,7-trimethylbicyclo[3.1.1]hept-2-en-4-one	C_10_H_14_O	97.45	-	11.6 ± 1.37	-	Monoterpenoids
20	14.746	-	(1R,5S)-2-methyl-5-prop-1-en-2-ylcyclohex-2-en-1-ol	C_10_H_16_O	96.49	-	3.86 ± 0.35	-	Monoterpenoids
21	15.126	-	2-methyl-5-prop-1-en-2-ylcyclohex-2-en-1-ol	C_10_H_16_O	89.83	-	0.08 ± 0.04	-	Monoterpenoids
22	15.267	-	2-methoxy-4-methyl-1-propan-2-ylbenzene	C_11_H_16_O	88.87	-	0.02 ± 0.02	-	Ether
23	15.434	-	4-propan-2-ylbenzaldehyde	C_10_H_12_O	84.86	-	0.13 ± 0.05	-	Aldehyde
24	15.536	-	(5S)-2-methyl-5-prop-1-en-2-ylcyclohex-2-en-1-one	C_10_H_14_O	94.2	-	0.66 ± 0.09	-	Monoterpenoids
25	15.795	-	2,7,7-trimethylbicyclo[3.1.1]hept-2-en-4-one	C_10_H_14_O	85.38	-	1 ± 0.09	-	Monoterpenoids
26	15.989	-	[(1S,2R,5R)-6,6-dimethylbicyclo[3.1.1]hept-2-yl]methanol	C_10_H_18_O	86.84	-	0.06 ± 0.04	-	Alcohols
27	16.693	-	1,7,7-Trimethylbicyclo[2.2.1]heptan-2-ol acetate	C_12_H_20_O_2_	86.53	-	0.27 ± 0.07	-	Esters
**28**	**16.849**	**10.793**	**(2E,4Z)-deca-2,4-dienal**	**C_10_H_16_O**	**83.19**	**92.02**	**0.08 ± 0.05**	**0.12 ± 0.06**	**Aldehyde**
29	16.979	-	[(4S)-4-prop-1-en-2-yl-1-cyclohexenyl]methanol	C_10_H_16_O	81.6	-	0.62 ± 0.11	-	Monoterpenoids
30	17.315	-	1-(2-hydroxy-5-methylphenyl)ethanone	C_9_H_10_O_2_	83.2	-	0.02 ± 0.02	-	Ketones
31	17.359	-	4-(2,2,6-trimethyl-1-bicyclo[4.1.0]heptanyl)butan-2-one	C_14_H_24_O	82.36	-	0.03 ± 0.03	-	Ketones
**32**	**17.382**	**11.195**	**(2E,4E)-deca-2,4-dienal**	**C_10_H_16_O**	**92.36**	**-**	**0.16 ± 0.1**	**0.18 ± 0.09**	**Aldehyde**
**33**	**17.631**	**11.481**	**2-Hydroxy-4-methoxybenzaldehyde**	**C_8_H_8_O_3_**	**97.42**	**-**	**2.62 ± 0.74**	**91.74 ± 1.15**	**Aldehyde**
34	18.125	-	(3R,3aS,7S,8aS)-3,6,8,8-Tetramethyl-2,3,4,7,8,8a-hexahydro-1H-3a,7-methanoazulene	C_15_H_24_	80.4	-	0.07 ± 0.07	-	Sesquiterpenes
35	18.241	-	2-Methoxy-4-prop-2-enylphenol	C_10_H_12_O_2_	94.36	-	0.32 ± 0.32	-	Phenols
**36**	**18.661**	**12.302**	**Tricyclo[4.4.0.02,7]dec-3-ene,1,3-dimethyl-8-(1-methylethyl)-, stereoisomer**	**C_15_H_24_**	**83.88**	**-**	**0.43 ± 0.16**	**0.18 ± 0.18**	**Sesquiterpenes**
37	18.752	-	(4aR,8aS)-7-Isopropylidene-4a-methyl-1-methylene-decahydro-naphthalene	C_15_H_24_	84.81	-	0.14 ± 0.1	-	Sesquiterpenes
38	18.771	-	[1S-(1α,3aβ,4α,8aβ,9S*)]-decahydro-4,8,8-trimethyl-1,4-methanoazulene-9-methyl acetate	C_17_H_28_O_2_	80.84	-	0.17 ± 0.08	-	Esters
39	18.873	-	(1R,3aS,5aS,8aR)-1,3a,4,5a-Tetramethyl-1,2,3,3a,5a,6,7,8-octahydrocyclopenta[c]pentalene	C_15_H_24_	81.46	-	0.02 ± 0.02	-	Sesquiterpenes
40	18.934	-	1-ethenyl-1-methyl-2,4-di(prop-1-en-2-yl)cyclohexane	C_15_H_24_	81.93	-	0.02 ± 0.02	-	Sesquiterpenes
**41**	**19.453**	**13. 051**	**(1R,4E,9S)-4,11,11-trimethyl-8-methylidenebicyclo[7.2.0]undec-4-ene**	**C_15_H_24_**	**84**	**-**	**0.63 ± 0.18**	**0.15 ± 0.15**	**Sesquiterpenes**
42	19.653	-	(1S,2E,10R)-3,7,11,11-Tetramethylbicyclo[8.1.0]undeca-2,6-diene	C_15_H_24_	84.06	-	0.18 ± 0.11	-	Sesquiterpenes
**43**	**19.784**	**13.326**	**1-(2-Hydroxy-4-methoxyphenyl)ethanone**	**C_9_H_10_O_3_**	**95.59**	**-**	**0.5 ± 0.39**	**0.05 ± 0.05**	**Phenols**
44	20.011	-	1,3a-Ethano-3aH-indene, 1,2,3,6,7,7a-hexahydro-2,2,4,7a-tetramethyl-, [1R-(1α,3aα,7aα)]-	C_15_H_24_	91.19	-	0.49 ± 0.32	-	Sesquiterpenes
45	20.353	-	1,4-dimethyl-7-propan-2-ylidene-2,3,4,5,6,8-hexahydro-1H-azulene	C_15_H_24_	88.63	-	0.27 ± 0.12	-	Sesquiterpenes
46	20.538	-	1,4-dimethyl-7-prop-1-en-2-yl-1,2,3,3a,4,5,6,7-octahydroazulene	C_15_H_24_	87.79	-	0.77 ± 0.45	-	Sesquiterpenes
47	20.719	-	(1S,4aS,8aR)-4,7-dimethyl-1-(propan-2-yl)-1,2,4a,5,6,8a-hexahydronaphthalene	C_15_H_24_	84.66	-	0.48 ± 0.17	-	Sesquiterpenes
**48**	**21.04**	**14.593**	**(1S,8aR)-4,7-Dimethyl-1-(propan-2-yl)-1,2,3,5,6,8a-hexahydronaphthalene**	**C_15_H_24_**	**83.12**	**-**	**1.57 ± 0.42**	**0.19 ± 0.19**	**Sesquiterpenes**
49	21.26	-	(1R,4aR,4bS,7R,10aR)-1,4a,7-Trimethyl-7-vinyl-1,2,3,4,4a,4b,5,6,7,9,10,10a-dodecahydrophenanthrene-1-carbaldehyde	C_20_H_30_O	89.3	-	4.95 ± 1.79	-	Sandaracopimaral
50	21.41	-	ent-Kaur-16-en-19-al	C_20_H_30_O	89.14	-	1.2 ± 1.16	-	Diterpene
51	21.914	-	(1aR,4aR,7S,7aR,7bR)-1,1,7-Trimethyl-4-methylenedecahydro-1H-cyclopropa[e]azulen-7-ol	C_15_H_24_O	93.04	-	1.31 ± 0.2	-	Sesquiterpenes
52	22.03	-	(1R,4R,6R,10S)-4,12,12-Trimethyl-9-methylene-5-oxatricyclo[8.2.0.0]dodecane	C_15_H_24_O	82.04	-	1.28 ± 0.25	-	Sesquiterpenes
53	22.742	-	(1aR,7S,7aS,7bR)-1,1,4,7-Tetramethyl-1a,2,3,5,6,7,7a,7b-octahydro-1H-cyclopropa[e]azulen-7-ol	C_15_H_24_O	86.65	-	0.14 ± 0.1	-	Sesquiterpenes
54	22.955	-	2-[(3S,5R,8S)-3,8-dimethyl-1,2,3,4,5,6,7,8-octahydroazulen-5-yl]propan-2-ol	C_15_H_26_O	86.34	-	0.19 ± 0.19	-	Sesquiterpenes
55	23.179	-	2-[(2R,4aR,8aS)-4a-methyl-8-methylidene-1,2,3,4,5,6,7,8a-octahydronaphthalen-2-yl]propan-2-ol	C_15_H_26_O	91.92	-	0.91 ± 0.44	-	Sesquiterpenes
**56**	**23.315**	**16.639**	**(4aS,8aR)-3,8a-Dimethyl-5-methylene-4,4a,5,6,7,8,8a,9-octahydronaphtho[2,3-b]furan**	**C_15_H_20_O**	**86.49**	**-**	**0.29 ± 0.22**	**0.41 ± 0.14**	**Sesquiterpenes**
57	24.173	-	(Z)-octadec-9-en-1-ol	C_18_H_36_O	84.2	-	0.1 ± 0.08	-	Alcohols
58	25.548	-	5-(5,5,8a-trimethyl-2-methylidene-3,4,4a,6,7,8-hexahydro-1H-naphthalen-1-yl)-3-methylpent-1-en-3-ol	C_20_H_34_O	87.84	-	4.9 ± 0.78	-	Diterpene
59	26.726	-	(3R)-5-[(1S,4aR,5S,8aR)-5-(Hydroxymethyl)-5,8a-dimethyl-2-methylenedecahydro-1-naphthalenyl]-3-methyl-1-penten-3-ol	C_20_H_34_O_2_	81.12	-	0.26 ± 0.26	-	Diterpene
60	29.794	-	2-ethenyl-2,4b-dimethyl-8-methylidene-3,4,4a,5,6,7,8a,9-octahydro-1H-phenanthrene	C_19_H_28_	82.09	-	0.33 ± 0.23	-	Alkanes
61	-	9.182	(E)-2-ethylhex-2-enal	C_8_H_14_O	-	83.58	-	0.04 ± 0.04	Aldehyde
62	-	9.701	3,4,5-trimethyloxolan-2-one	C_7_H_12_O_2_	-	87.35	-	0.5 ± 0.1	Ketones
63	-	10.197	4-Hydroxy-3-methylbenzaldehyde	C_8_H_8_O_2_	-	84.32	-	0.01 ± 0.01	Aldehyde
64	-	12.579	3-Hydroxy-4-methoxybenzaldehyde	C_8_H_8_O_3_	-	86.95	-	0.13 ± 0.04	Alkanes
65	-	13.187	1-ethenyl-1-methyl-4-propan-2-ylidene-2-prop-1-en-2-ylcyclohexane	C_15_H_24_	-	85.61	-	0.05 ± 0.02	Sesquiterpenes
66	-	13.474	5-formyl-2-methoxyphenyl acetate	C_10_H_10_O_4_	-	91.55	-	2.1 ± 0.37	Monoterpenoids
67	-	13.589	(1E,4E,8E)-2,6,6,9-tetramethylcycloundeca-1,4,8-triene	C_15_H_24_	-	87.39	-	0.05 ± 0.05	Sesquiterpenes
68	-	13.688	methyl 2-hydroxy-4-methoxybenzoate	C_9_H_10_O_4_	-	97.04	-	0.53 ± 0.06	Esters
69	-	14.096	(3R,4aS,8aR)-8a-methyl-5-methylidene-3-prop-1-en-2-yl-1,2,3,4,4a,6,7,8-octahydronaphthalene	C_15_H_24_	-	85.68	-	0.12 ± 0.04	Sesquiterpenes
70	-	14.193	(3S,3aR,3bR,4S,7R,7aR)-4-Isopropyl-3,7-dimethyloctahydro-1H-cyclopenta[1,3]cyclopropa[1,2]benzen-3-ol	C_15_H_26_O	-	92.71	-	0.06 ± 0.06	Sesquiterpenes
71	-	14.282	2,4-ditert-butylphenol	C_14_H_22_O	-	93.44	-	0.58 ± 0.13	Phenols
72	-	14.507	(3R,3aR,3bR,4S,7R,7aR)-4-Isopropyl-3,7-dimethyloctahydro-1H-cyclopenta[1,3]cyclopropa[1,2]benzen-3-ol	C_15_H_26_O	-	92.58	-	0.27 ± 0.27	Sesquiterpenes
73	-	14.834	1,2,3,5,6,7,8,8a-octahydro-1,8a-dimethyl-7-(1-methylethenyl)-,[1R-(1a,7b,8aa)]-Naphthalene	C_15_H_24_	-	85.36	-	0.13 ± 0.05	Sesquiterpenes
74	-	14.801	(4aR,8aR)-4a,8-dimethyl-2-(1-methylethylidene)-1,2,3,4,4a,5,6,8a-octahydronaphthalene	C_15_H_24_	-	81.81	-	0.03 ± 0.02	Sesquiterpenes
75	-	16.354	(1S,4S,4aR,8aR)-1,6-dimethyl-4-propan-2-yl-3,4,4a,7,8,8a-hexahydro-2H-naphthalen-1-ol	C_15_H_26_O	-	93.83	-	0.25 ± 0.25	Sesquiterpenes
76	-	17.521	7-methoxychromen-2-one	C_10_H_8_O_3_	-	86.21	-	0.01 ± 0.01	Phenylpropanoids
77	-	18.672	hexadecanal	C_16_H_32_O	-	92.5	-	0.06 ± 0.06	Aldehyde
78	-	18.932	(9Z,12Z)-octadeca-9,12-dienoic acid	C_18_H_32_O_2_	-	81.53	-	0.35 ± 0.35	Organic acids
79	-	19.113	(E)-octadec-9-enoic acid	C_18_H_34_O_2_	-	80.34	-	0.03 ± 0.03	Organic acids
80	-	19.615	O1-cyclohexyl O2-(2-methylpropyl) benzene-1,2-dicarboxylate	C_18_H_24_O_4_	-	83.82	-	0.01 ± 0.01	Esters
81	-	20.321	(3S,3aS,6S,7S,7aS)-7-Isopropenyl-3,6-dimethyl-6-vinyl-hexahydro-benzofuran-2-one	C_15_H_22_O_2_	-	84.77	-	0.08 ± 0.08	Esters
82	-	21.551	O2-butyl O1-cyclohexyl benzene-1,2-dicarboxylate	C_18_H_24_O_4_	-	87.88	-	0.05 ± 0.05	Esters

-, not detected. Co-contained constituents are indicated in bold type.

## Data Availability

The study did not report any data.

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
