# Peer review of "Comparative Analysis of Acanthopanacis Cortex and Periplocae Cortex Using an Electronic Nose and Gas Chromatography–Mass Spectrometry Coupled with Multivariate Statistical Analysis"

_molecules, 2022, doi:10.3390/molecules27248964_

Round 1
Reviewer 1 Report (Previous Reviewer 1)
The authors presented corrections to their manuscript and answers to the remarks in the previous version.
The authors present the distribution of data points in the principal component space. I expressed concerns about the number of points in the charts as it is not equal to the number of performed measurements. The authors explained that for these charts they used averages over several measurements and cited other papers. I think it is a rather unusual way of presentation of such data. According to the authors' knowledge, a similar method of averaging was used in other papers. I verified the references cited in the answers to the reviewers and in my opinion, the authors are wrong and the cited papers present individual measurement points. For example, in cited by the authors' manuscript T. Feng et al. / Sensors and Actuators B 160 (2011) 964– 973, figures 2 - 4. In another cited by the authors' paper J Food Process Preserv. 2019;43:e14095, Figure 3, one can also notice that the individual data points are plotted.
In my opinion, the presentation of the averages instead of raw data may lead to a misleading impression of this figure. It may happen, that when raw data will be presented there may be not so distinct separation between measurements of two species and even some overlap of measurements.
I expressed my doubts concerning the results of the analysis of GCMS data by the OPLS-DA model, performed to find chemical markers allowing to differentiate between the studied samples.
As a result of such analysis, the authors received a list of 24 chemical markers, which can be used to differentiate between two studied species. I expressed some concerns regarding this list using arguments from simple univariate analysis (which of course I cannot perform in detail as I have no detailed data). The authors claim that multivariate analysis is more appropriate for their task.
In my opinion, the authors should provide more interpretation of the result, not just a coloring of Figure 5. For example, does each listed chemical component can be used as a marker allowing differentiation? The presence of a component in a measured sample is a marker of which herb species? What concentration of a component in a sample is a marker of a herb species? Are there "better" markers that other and magnitude of VIP (fig. 5) can represent such a quality of differentiation, and if so does the magnitude gives any clue about the possibility to differentiate of samples or only the order is important? Or maybe, since the used method is multivariate only the composition of some (all?) of this list of components can be used used as markers to differentiate between species. If more than one chemical component is needed as a marker to differentiate between herbs species, how many of them?
The authors should provide relevant references to other works in which the same method was used for the identification of chemical markers.
Author Response
Please see the attachment.

Reviewer 2 Report (Previous Reviewer 3)
The revised manuscript is well revised. Before submitting final version of the manuscript, please confirm the following points.
1. Table 1, Column 1, "N." should be "No.".
2. Table 1, Column 2, tR should be "Retention time".
3. Table 1, Column 5, "Compatibility".
4. Table S1, "Sample name" and "Batch code".
5. References, Journal titles should be correctly described with abbreviations.
6. 2.4.2, The abbreviation of GC-MS should be defined in Introduction.
7. Table 1, title, VOC should be defined in section 2.5
8. Fig. 1, title abbreviation of PCA was already defined.
9. Fig. 2, title, abbreviation of OPLS-DA was already defined.
10. Fig. 3, title, abbreviation of RDA was already defined.
11. Fig. 5, title, abbreviation of VOC was already defined.
Round 2
Reviewer 1 Report (Previous Reviewer 1)
the authors improved the manuscript
This manuscript is a resubmission of an earlier submission. The following is a list of the peer review reports and author responses from that submission.
Round 1
Reviewer 1 Report
The authors collect all charts to only two figures. Such a practice makes the manuscript more difficult to read compared to the case if there would be separate figures. The practice to join subfigures to one figure may be justified if the same property is presented or if it helps the reader to compare presented values more easily, but it is not the case.
Also figures captions should be more precise and sufficiently explain the presented figures.
The authors present some figures, which are not mentioned in the text and it is not easy to understand what they present and why the authors included these figures. For example Figure 1 has subfigures a-e and only, a,b, and e are mentioned.
It is not sure what are dots X1-X12 and W1-W16 are presented in Figure 1. One can guess that they represent the samples measured by electronic nose and a list of these samples are provided in supplementary materials. However, as the authors mentioned in the manuscript, each sample was measured 3 times. So the figure should contain much more dots. It is very improbable that all the measurements of the herb sample gave the same results. Maybe the authors used average value, but it is just my guess and in my opinion, such averaging cannot be easily justified as the sensor's response is nonlinear.
Even more doubts I have about the results presented in Figure 2. There the authors plot the results of GCMS analysis and also there are separate dots for X1-X12 and W1-W16. That suggests, that each of the biological samples was measured. That is in contraction in the caption of Table 1, where the authors state in total that only 3 measuments of W and X were performed.
If there were only 3 measurements, the authors should specify which biological samples were used and why the limited their studies for such a choice. But that approach makes figures 2 a, and b impossible to prepare as the authors would had not obtain such detailed data.
However, if the authors performed detailed GCMS measurements of each biological sample, then these results should be reflected in the analysis. For example, the authors present in Table 1 a list of chemical components detected by the GCMS method. The relative content of each component is presented together with the standard deviation and looking at these numbers it seems that some of the components were not detected in all samples of a given category. That suggests, that they probably cannot be considered as a general property of studied species. Do the authors perform studies of 12 (or 16) biological samples and can distinguish between components specific to all samples of X or all samples of W?
The results presented in Table 1 demonstrate that sample X is composed of 91.75% of one chemical component. In the discussion section, the authors cite Ref.[24]. Unfortunately I have no access to that position so I could not verify this citaiton. However, that result seems in contradiction with the results presented in Ref. [21] (Molecules 2019, 24, 3621; doi:10.3390/molecules24193621) where multiple chemical components were reported for that herb, with no such dominance of this compound, and also there are found important differences between herbs origins.
The authors present a selection of chemical components that may be used as markers allowing differentiation between X and W. However, as I mentioned they have not demonstrated that these components are present in all samples of a given category. For some of them, the content is very low, often below 1%. I looked at the table from the previous version of the manuscript, and it seems that indeed some of the candidates for markers were not present in all samples of the given category, for example, N25, N24, N71, and N64. I verified it, by comparing, the components with the same retention time, as the numbering is changed. Some other chemical components, that seem to be present in all samples of only one category were not selected by the used algorithm. That observation gives reasonable doubts if the applied authors' method of selection of chemical markers is justified, or if it is correctly applied.
Since the goal of the authors is to demonstrate differences between two herb species they should report the properties that could be generalized regardless of the origin of the samples. The analysis presented by the authors is either not sufficiently described or does not answer to that goal.
On page 7 the authors present the contribution to the variance of three principal components: 54.5%, 29.0%, and 16.5%. As the authors claim, they are the transformation of data from 10 sensors. The sum of the contribution of these first three components is exactly 100%. That means that these 3 new variables contain all information from the original 10 variables. In my opinion, it is rather an error in calculations than a real result.
Reviewer 2 Report
The authors resubmitted their manuscript. The new version has been adequately revised according to my previous comments. Therefore, I recommend its acceptance for publication.
Reviewer 3 Report
The manuscript describes analysis of Chinese Herbal Medicine oil samples by GC-MS and an electronic nose. The results obtained by E-nose were further analysed by pattern analysis.
Unfortunately, the content is flimsy, and I can't find enough novelty of the work. Furthermore, I think that this kind of analysis of regional plants is not suitable for published in an international journal without outstanding novelty.
Therefore, I can't recommend acceptance of this manuscript.
Specific comments
1. Introduction, Line 2, should be "in clinical application for a long time in China."
2. Introduction, abbreviation that once defined must be used. i.g. MOS and PCA.
3. 2.4.2, A space must be inserted between a number and a unit. (column information)
4. 2.4.2, Sample injection volume and split ratio must be described.
5. 3.3, title, "Results of" is not needed. Section 3 is result section.
6. 4.1, title, I don't that the title is appropriate.
7. 4.1, I think that the contents described in section 4.1 is very flimsy.
8. 4.5, title, I don't think the title is appropriate.